

# Impact of fructooligosaccharides on gut microbiota composition and metabolite production: implications for childhood obesity

CanFeng Mo[1,*], Shan Zhou[2,*], Zhi Du[3] and Xiangxin Huang[4]

[1] Department of Pharmacy, Affiliated Hangzhou First People's Hospital, School of Medicine, Westlake University, Hangzhou, Zhejiang, China

[2] Department of Ultrasound, The Affiliated Huaian No. 1 People's Hospital of Nanjing Medical University, Huaian, China

[3] Department of Pharmacy, Children's Hospital, Zhejiang University School of Medicine, National Clinical Research Center for Child Health, Hangzhou, Zhejiang, China

[4] Department of Clinic Trial, Affiliated Hangzhou First People's Hospital, School of Medicine, Westlake University, Hangzhou, Zhejiang, China

[*] These authors contributed equally to this work.

Corresponding author
Xiangxin Huang,
huangxx_123@163.com

## ABSTRACT

**Background**. Contemporary dietary habits in children have been linked to various health issues, particularly the increasing prevalence of childhood obesity. However, the complex relationship between children's diets, gut microbiota, and health outcomes remains incompletely understood. This study investigates the effects of fructooligosaccharides (FOS) on gut microbiota composition and metabolic processes in children, and explores their potential impact on pediatric health outcomes such as obesity and metabolic disorders.

**Methods**. Fecal samples from 39 children (19 boys, 20 girls) aged 6–15 were subjected to *in vitro* fermentation with or without FOS supplementation. Bacterial composition, short-chain fatty acid (SCFA) production, and gas generation were analyzed. Potential biomarkers and associations between gut microbiota, metabolites, and metabolic pathways were identified using Random Forest algorithms and the MetOrigin cloud platform.

**Results**. FOS supplementation significantly altered the $\beta$-diversity of the gut microbiota, increasing the abundance of *Bifidobacterium* and *Lactobacillus*, while decreasing *Escherichia-Shigella* and *Bacteroides*. FOS also led to a significant increase in SCFA levels, particularly acetic acid, which correlated positively with *Bifidobacterium* and negatively with *Streptococcus*. Conversely, gas production ($NH_3$, $H_2$, and $H_2S$) decreased significantly and showed a positive correlation with *Escherichia-Shigella* and a negative correlation with *Bifidobacterium*.

**Conclusions**. This study highlights changes in microbial structure, metabolite production, potential biomarkers, and altered metabolic pathways following FOS intervention. These findings provide valuable insights into the complex relationship between diet and gut microbiota in obese children and suggest that dietary interventions may influence pediatric health through modulation of the gut microbiome.

# INTRODUCTION

Contemporary dietary trends among children are increasingly contributing to the alarming rise in childhood obesity, accompanied by a surge in health issues such as type 2 diabetes, cardiovascular diseases, and various metabolic disorders (*Zhang et al., 2015*). The pervasive consumption of highly processed and sugary foods not only intensifies weight gain but also plays a significant role in exacerbating these health complications. Despite their evident impact, the intricate dynamics between children's dietary habits and their long-term health effects remain poorly understood and under-researched.

Diet significantly influences the delicate balance of gut microbiota in children. The shift towards high consumption of processed and sugary foods, combined with a reduction in fiber-rich and nutrient-dense alternatives, disrupts gut microbial composition and diversity (*Tan et al., 2022*). This disruption can decrease beneficial bacteria while increasing potentially harmful ones, leading to effects that extend beyond the digestive system and impact overall health (*Taverno Ross et al., 2020*; *Zhang et al., 2020*). Such dietary shifts correlate with heightened risks of obesity, metabolic disorders, and impaired immune functions, underlining the critical interplay between diet and gut microbiota (*Liu et al., 2023*; *Sonnenburg & Backhed, 2016*). In our prior study, stachyose was identified as a factor influencing microbial community and metabolites, underscoring the significance of dietary interventions that target the gut microbiota and metabolic function in the management of childhood obesity (*Pi et al., 2024a*). Although both stachyose and fructooligosaccharides (FOS) are oligosaccharides, their effects on the gut microbiota and metabolic functions are distinct due to differences in their chemical structure and fermentation profiles (*Li, Lu & Yang, 2017*; *Zhang et al., 2022a*). Further investigation is required to gain a deeper understanding of the specific effects of each oligosaccharide on the composition and metabolic pathways of the gut microbiota in obese children.

FOS are vital dietary fibers that enhance gut microbiota. As non-digestible carbohydrates, FOS reach the colon intact and promote the growth of beneficial bacteria such as *Bifidobacterium* and *Lactobacillus* (*Benito et al., 2021*; *Bi et al., 2022*; *Dong et al., 2023*; *Mao et al., 2019*). The fermentation of FOS yields short-chain fatty acids (SCFAs) like acetic, propionic, and butyric acids, fostering an acidic gut environment that inhibits the proliferation of pathogens (*Ayimbila et al., 2022*; *Charoensiddhi, Chanput & Sae-Tan, 2022*; *Roupar et al., 2022*). The modulation of gut microbiota by FOS is associated with improved intestinal health, better nutrient absorption, and potential systemic benefits (*Kumar, Rani & Datt, 2020*).

The purpose of this study is to examine the influence of FOS on microbial and metabolic processes in obese children, with a specific emphasis on its potential impact on pediatric health concerns such as obesity and metabolic disorders. By employing *in vitro* simulated fermentation, which replicates the conditions of the gastrointestinal tract outside the body,

this research provides a detailed examination of the interactions between FOS and the gut microbiota, offering controlled insights into the potential benefits of FOS in pediatric health.

## MATERIALS & METHODS

### Materials and reagents

FOS with a purity of $\geq 95\%$ and moisture content of $\leq 5\%$ (Quantum Hi-Tech Biological Co., Ltd.). L-cysteine, bile salt, heme, and yeast extract were procured from Sigma-Aldrich, USA. Sodium chloride, dipotassium hydrogen phosphate, potassium dihydrogen phosphate, magnesium sulfate, metaphosphoric acid, calcium chloride, phosphate-buffered saline (PBS), and crotonic acid were bought from Sangon Biotech (Shanghai) Co., Ltd, China. The yeast extract-casitone-fatty acid (YCFA) medium was sourced from Dingguochangsheng Biotechnology (Beijing) Co., Ltd, China.

### Collection of fresh fecal samples

The study recruited a total of 39 participants, aged 6-15, were recruited for the study, 19 boys and 20 girls. Body mass index (BMI) was calculated by dividing the participants' weight (in kilograms) by their height (in meters), providing an assessment of their weight status. BMI-for-age $Z$-scores or each age group were determined according to World Health Organization (WHO) guidelines and classified according to Chinese standards (*Song et al., 2014*). All participants were identified as overweight or obese and had no prior history of digestive disorders, nor had they used antibiotics, probiotics, or prebiotics in the month preceding the study. A dietary recall interview was conducted with participants, who were asked to report all foods and beverages consumed in the past 24 h. Multiple recalls were conducted over the course of a week to capture changes in dietary intake. The results indicated a tendency towards the consumption of highly processed and sugary foods, aligning with the recommendations outlined in the 2016 Dietary Guidelines for Chinese Residents (*Wang et al., 2016*). All participants resided in Hangzhou, Zhejiang Province. Ethical approval for the study protocol was obtained from the Ethical Committee of Hangzhou Centers for Disease Control and Prevention (No. 202047). Informed consent was obtained in the form of written consent signed by all participants or their parents/legal guardians prior to their participation in the study.

### Treatment of fresh fecal samples

Fresh fecal samples were collected in a 30 mL sterile sample box (91 mm $\times$ $\Phi$24 mm) provided by BioRise Co., Ltd. (Shanghai, China). The fresh fecal samples were collected between August 10, 2021, and November 16, 2021, with collection times during the day (between 8:00 AM and 5:00 PM) to minimize variability due to circadian effects on the gut microbiota. Immediately after defecation, a minimum of 3 g intermediate feces were selected, with information regarding the volunteer included. To minimize any potential degradation or alterations in microbial composition, at least 3 g of partially processed fecal material, containing minimal undigested food residue and exposed to air for only a brief period post-defecation, was transferred to an anaerobic jar and stored at 4 °C immediately after defecation.

## Isolation of gut microbiota

Three 1.5-mL sterile centrifuge tubes were obtained, with each tube containing 0.2 g of fresh fecal matter. The tubes were stored as backup raw fecal samples at $-80\,°C$. Next, eight mL of sterile PBS and 0.8 g of fresh fecal sample were added to 10 mL sterile centrifuge tubes under anaerobic conditions. These tubes were sealed with tape and placed on a shaker for thorough mixing. The resulting supernatant was then filtered to prepare a 10% suspension of gut microbiota (*Pi et al., 2024b*).

## Preparation of medium

To prepare the YCFA medium, a mixture was formulated consisting of the following components: 4.5 g/L yeast extract, 3.0 g/L peptone, 3.0 g/L tryptone, 0.4 g/L bile salt, 0.8 g/L cysteine hydrochloride, 2.5 g/L potassium chloride, 4.5 g/L sodium chloride, 0.45 g/L magnesium chloride, 0.2 g/L calcium chloride, 0.4 g/L potassium dihydrogen phosphate, 1.0 mL Tween 80, 2.0 mL trace element solution, and 1.0 mL resazurin. After the dissolution and boiling processes, nitrogen gas was introduced to the liquid medium to maintain anaerobic conditions. Under these conditions, 4.5 mL of the medium was transferred into 10 mL vials using a peristaltic pump (Longer Co., Ltd., Baoding, China). The vials were sealed and autoclaved using a heat-pressure steam sterilizer obtained from Shen An Co. (Shanghai, China). Sterilization was carried out at 115 °C and 101 kPa for 15 min (*Chen et al., 2021*). In the FOS group, 0.8 g/100 mL of FOS was incorporated into the YCFA medium, whereas the YCFA group received no FOS supplementation.

## *In vitro* fermentation of gut microbiota

Under anaerobic conditions, the 10% suspensions of gut microbiota were divided into two portions: 0.5 mL each and inoculated into the culture media of the YCFA and FOS groups, respectively, using a disposable sterile syringe. Three parallel batches of each medium were prepared, gently shaken to ensure thorough mixing, and incubated for 24 h at 37 °C in a constant-temperature incubator (Biobase, China).

## 16S rRNA sequencing of gut microbiota

16S rRNA sequencing was used to analyze the differences in gut microbiota communities following simulated *in vitro* fermentation. Genomic DNA was extracted from the samples using the FastDNA® Spin Kit for Soil (MP Biomedicals), according to the manufacturer's instructions. The V3–V4 hypervariable regions of the bacterial 16S rRNA gene were amplified using the universal primer pair 341F and 806R, as previously described (*Pi et al., 2024a*). Pair-end sequencing was performed on the purified amplicons using the NovaSeq PE250 platform (Majorbio Bio-pharm Technology Co., Shanghai, China).

After quality control, the DADA2 plug-in in QIIME2 was employed to denoise the optimized sequences, generating amplicon sequence variants (ASVs). To mitigate the influence of sequencing depth on alpha and beta diversity analyses, all samples were rarefied to 54,000 sequences. The $\alpha$-diversity was assessed using the ACE, Chao, and Shannon indices, calculated with Mothur (v1.30.1). For taxonomic classification of ASVs, a Naive Bayes classifier trained on the SILVA database (version 138) was applied within the QIIME2 framework. Sequencing data from both fermentation and raw fecal samples

were deposited in the National Center for Biotechnology Information (NCBI) Short Read Archive (SRA) under accession number PRJNA1201739.

## Measurement of SCFAs

After fermentation, the samples underwent centrifugation to separate the supernatant. This supernatant was then acidified for 24 h using a mixture of crotonic acid and metaphosphoric acid. Following acidification, the supernatant was filtered and centrifuged through a 0.22 µm aqueous microporous membrane (Millipore Express, Germany). The resulting filtered solution was transferred into a sample vial, with a volume of 150 µL. The sample was prepared for aging, and the gas chromatography (GC-2010 Plus; Shimadzu, Kyoto, Japan) was configured accordingly. The column temperature was programmed to rise from 80 °C to 190 °C at a rate of 10 °C/min, then to 240 °C at 40 °C/min, where it was held for 5 min. Both the flame ionization detector and the gasification chamber were set to 240 °C. The carrier gases used were nitrogen (20 mL/min), hydrogen (40 mL/min), and air (400 mL/min). Data were recorded, with trans-2-butenoic acid acting as the internal standard. Gas chromatography with a thin-film capillary column (DB-FFAP, 30 m × 0.32 mm × 0.50 µm, Agilent Technologies) was utilized to analyze the composition of short-chain fatty acids (SCFAs) in the culture filtrates, including acetic acid (Ace), propionic acid (Pro), butyric acid (But), isobutyric acid (Isob), pentanoic acid (Pen), and isovaleric acid (Isov) (*Pi et al., 2022*).

## Measurement of gases

Following a 24-hour fermentation period, the fermentation vials were removed and cooled to room temperature (25 °C). The composition and volume of the gases were automatically analyzed and recorded using a gas analyzer (Empaer, China) with five highly sensitive gas sensors, in accordance with a previously described (*Ye et al., 2022*). The concentrations of the five gases, including methane ($CH_4$), hydrogen sulfide ($H_2S$), ammonia ($NH_3$), carbon dioxide ($CO_2$), and hydrogen ($H_2$), were recorded at their peak values. Subsequent testing occurred when the values of all five gases dropped to zero.

## Data analysis

The dataset was expressed as the mean and standard error of the mean (SEM) and analyzed statistically using SPSS version 23.0 (IBM Corp., Armonk, NY, USA). Differences in $\alpha$-diversity, as measured by the ACE, Chao, and Shannon indices, were assessed using one-way ANOVA followed by Tukey–Kramer *post hoc* tests based on the ASV table. To evaluate structural variation in $\beta$-diversity at the genus level, principal coordinate analysis (PCoA) and non-metric multidimensional scaling (NMDS) were performed using the PERMANOVA (Permutational Multivariate Analysis of Variance) based on Bray-Curtis dissimilarity. The count data were converted to relative abundances prior to subsequent statistical analyses to ensure comparability across samples with varying sequencing depths. Graphical representations of SCFAs and gas profiles were generated with GraphPad Prism 8.0.3 (GraphPad Software, San Diego, CA, USA). The Spearman correlation coefficient was calculated to create a correlational heatmap. Additionally, linear discriminant analysis effect size (LEfSe) was used to identify differentially abundant microbial taxa between the

FOS and YCFA groups, with a threshold for the linear discriminant analysis (LDA) score set above 3.0. Random forest models were developed with base-10 logarithmic transformation for data normalization, and the area under the curve (AUC) was used for model assessment. These computational analyses and visualizations were facilitated by the Majorbio Cloud Platform (https://www.majorbio.com). Functional predictions were carried out using the MetOrigin Cloud Platform (https://metorigin.met-bioinformatics.cn), which enables functional annotation and enrichment analysis based on metabolomics data.

## RESULTS

### Exploring microbial diversity and composition post-fermentation

The $\alpha$-diversity indices (ACE, Chao, and Shannon indices) did not show any statistically significant differences between the YCFA and FOS groups (Figs. 1A–1C), suggesting a consistent community diversity across both groups. In contrast, $\beta$-diversity analyses, including NMDS and PCoA, revealed a notable distinction in genus-level bacterial community structures between the FOS and YCFA groups ($p < 0.001$; Figs 1D, 1E).

Moreover, the bar plots elucidate the variations in the relative abundance of genera across the samples (Fig. 2A). The count data were converted to relative abundances prior to statistical analysis to ensure comparability across samples with varying sequencing depths. In the FOS group, *Bifidobacterium* dominated, representing the highest relative abundance at 45.12%, followed by *Escherichia-Shigella* (15.08%), *Lactobacillus* (7.14%), and *Bacteroides* (5.12%) (Fig. 2B). In contrast, the YCFA group featured *Escherichia-Shigella* as the predominant genus, accounting for a relative abundance of 49.64%, followed by *Bacteroides* (10.36%), *Bifidobacterium* (5.73%), and *Phascolarctobacterium* (3.47%) (Fig. 2C).

To identify specific bacterial taxa with differential enrichment between the FOS and YCFA groups, LEfSe analysis was conducted. As illustrated in Fig. 2D, 31 genera exhibited significant differences (LDA > 3) in content between the FOS and YCFA groups. Notably, the FOS group showed significant enrichment in *Bifidobacterium*, *Lactobacillus*, *unclassified Bacilli*, and *Faecalibacterium*. In contrast, fecal samples from the YCFA group had a higher abundance of *Escherichia-Shigella*, *Bacteroides*, *unclassified Enterobacteriaceae*, and *Phascolarctobacterium*.

### SCFAs and gas production during fermentation

The total levels of SCFAs were significantly higher ($p < 0.0001$) in the FOS group compared to the YCFA group (Fig. 3A). This increase in total SCFAs in the FOS group was primarily attributed to acetic acid (Ace), which exhibited the highest yield among SCFAs and showed a significant increase (Fig. 3B). Conversely, propionic acid (Pro), isobutyric acid (Isob), and isovaleric acid (Isov) experienced significant decreases ($p < 0.01$) in the FOS group (Figs. 3D, 3E, and 3G). The correlation heatmap indicates a significantly strong positive correlation between the yield of acetate and *Bifidobacterium* (cor = 0.5, $p < 0.001$), as well as a significant negative correlation with *Streptococcus* (cor = $-0.5$, $p < 0.001$). Additionally, the yield of propionic acid (Pro) exhibited a significantly positive correlation

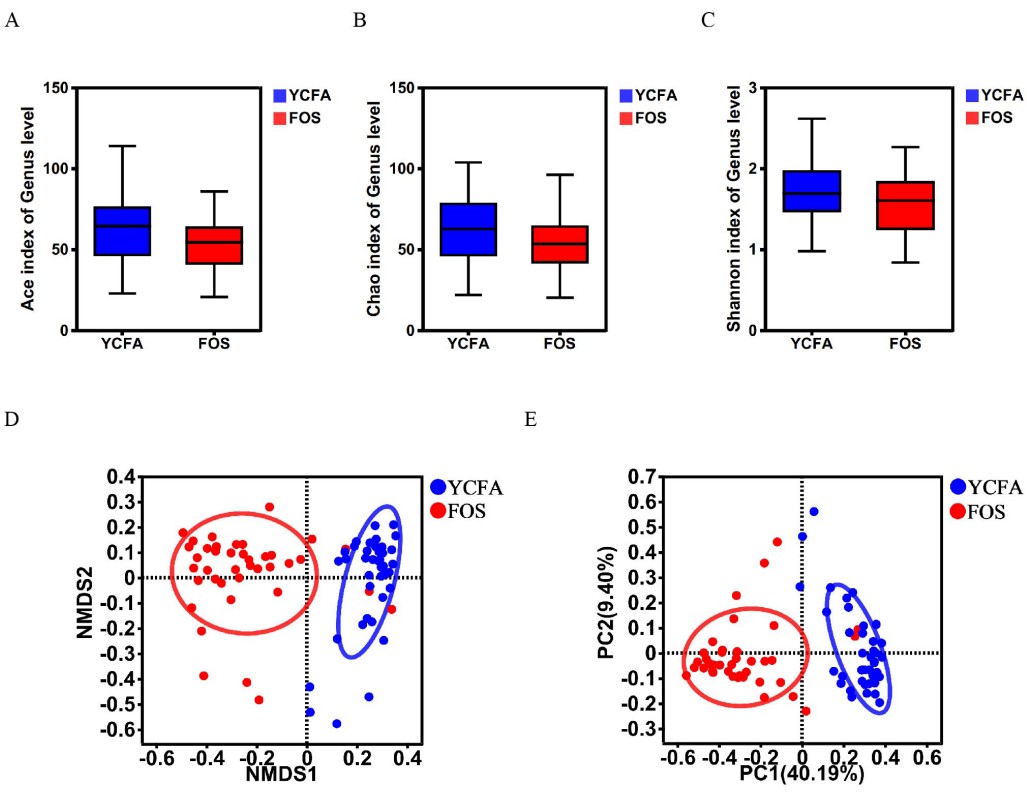

**Figure 1 Gut microbiota diversity post *in vitro* fermentation was evaluated.** $\alpha$-diversity indices, including the (A) ACE index ($p = 0.780$), (B) Chao1 index ($p = 0.220$), and (C) Shannon index ($p = 0.234$), utilizing the Wilcoxon rank-sum test to compare the FOS and YCFA groups. $\beta$-diversity analysis at the genus level, (D) NMDS ($p < 0.001$) and (E) PCoA ($p < 0.001$) were conducted using the Bray–Curtis metrics and ANOSIM to assess dissimilarity between the control group (YCFA group) and the treatment group (FOS group).

with *Escherichia-Shigella* (cor = 0.5, $p < 0.001$) and a significantly negative correlation with *Bifidobacterium* (cor = $-0.5$, $p < 0.001$) (Fig. 3H).

The gas content was significantly lower in the FOS group compared to the YCFA group ($p < 0.0001$, Fig. 3I). In both groups, the highest volume of gases was attributed to $H_2$, followed by $H_2S$, $CH_4$, and $NH_3$, with a smaller content of $CO_2$ (Figs. 3J–3N). The contents of $NH_3$, $H_2$, and $H_2S$ (Figs. 3L–3N, $p < 0.0001$) were significantly lower in the FOS group than in the YCFA group. In the correlation heatmap, it became evident that these three gases, which exhibited notable alterations in content, displayed significant positive correlations with *Escherichia-Shigella* (cor = 0.5, $p < 0.001$) and marked negative correlations with *Bifidobacterium* (cor = $-0.5$, $p < 0.001$). Additionally, a strong positive correlation was identified between the $NH_3$ content and *Bacteroides* (cor = 0.5, $p < 0.001$).

## Biomarker identification from gut microbiota and metabolites

To explore the intricate interplay among gut microbiota, metabolites, and their potential as biomarkers, a random forest algorithm was applied to determine the discriminative power of these factors in distinguishing between the FOS and YCFA groups. The top 28

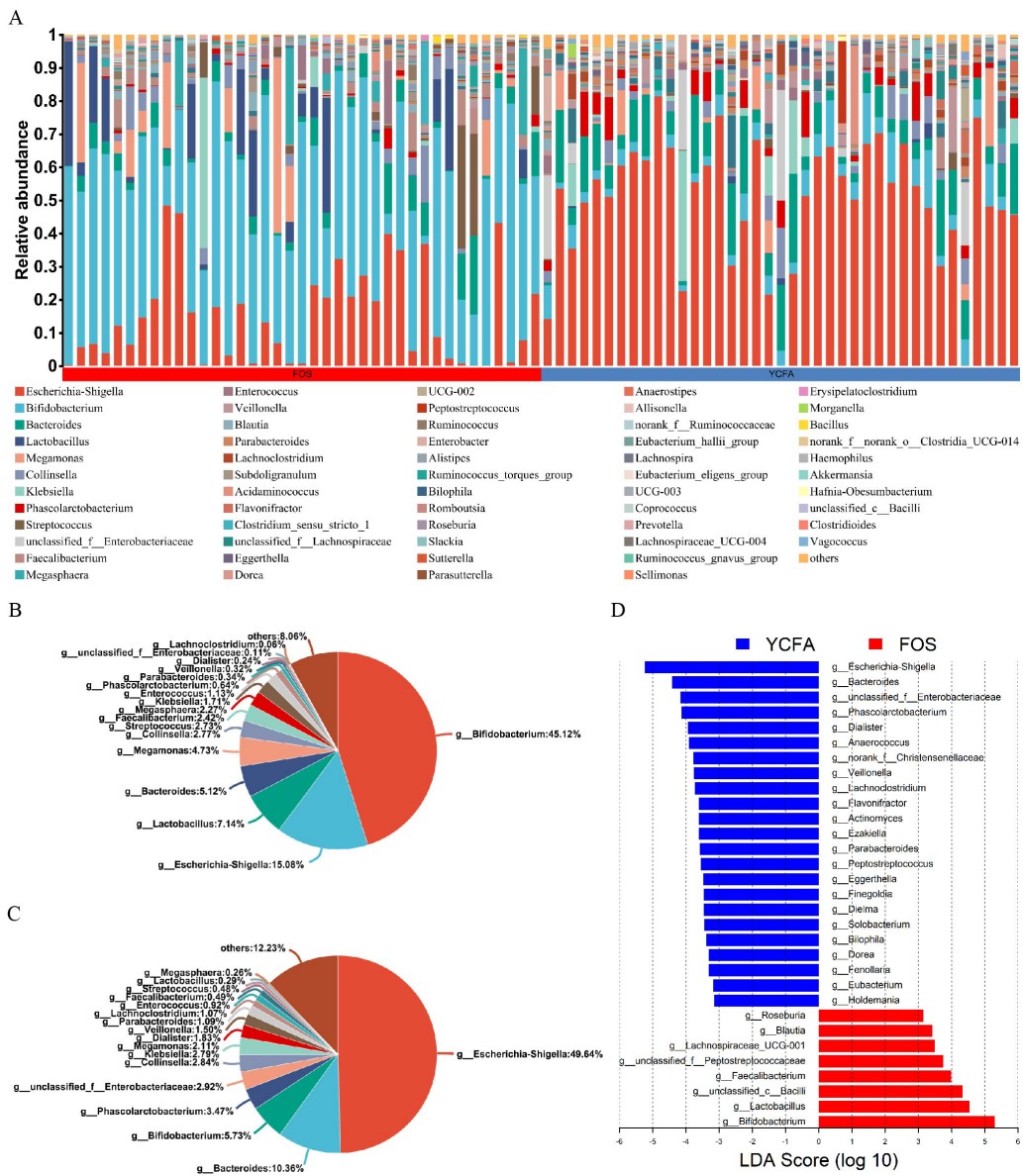

**Figure 2** **Gut microbiota composition post *in vitro* fermentation.** (A) Bar plot depicting genus-level bacterial community in the control group (YCFA group) and the treatment group (FOS group); pie charts illustrating genus-level bacterial community in the (B) FOS and (C) YCFA groups; (D) genus-level gut microbiota comparisons between the YCFA group and the FOS group analyzed with linear discriminant analysis effect size (LEfSe) (linear discriminant analysis, LDA) > 3, $p < 0.05$.

genera were strategically chosen based on their feature importance, constituting a crucial component of our validation queues (Fig. 4A). Subsequent analyses of specificity and sensitivity for these genera resulted in an area under the curve (AUC) of 0.77 (95% CI [0.66–0.88]) (Fig. 4B), highlighting their potential as indicators. Expanding our exploration to encompass various gut microbiota and metabolites, the top 21 variables of importance emerged as valuable biomarkers for differentiation (Fig. 4C). These variables showed an

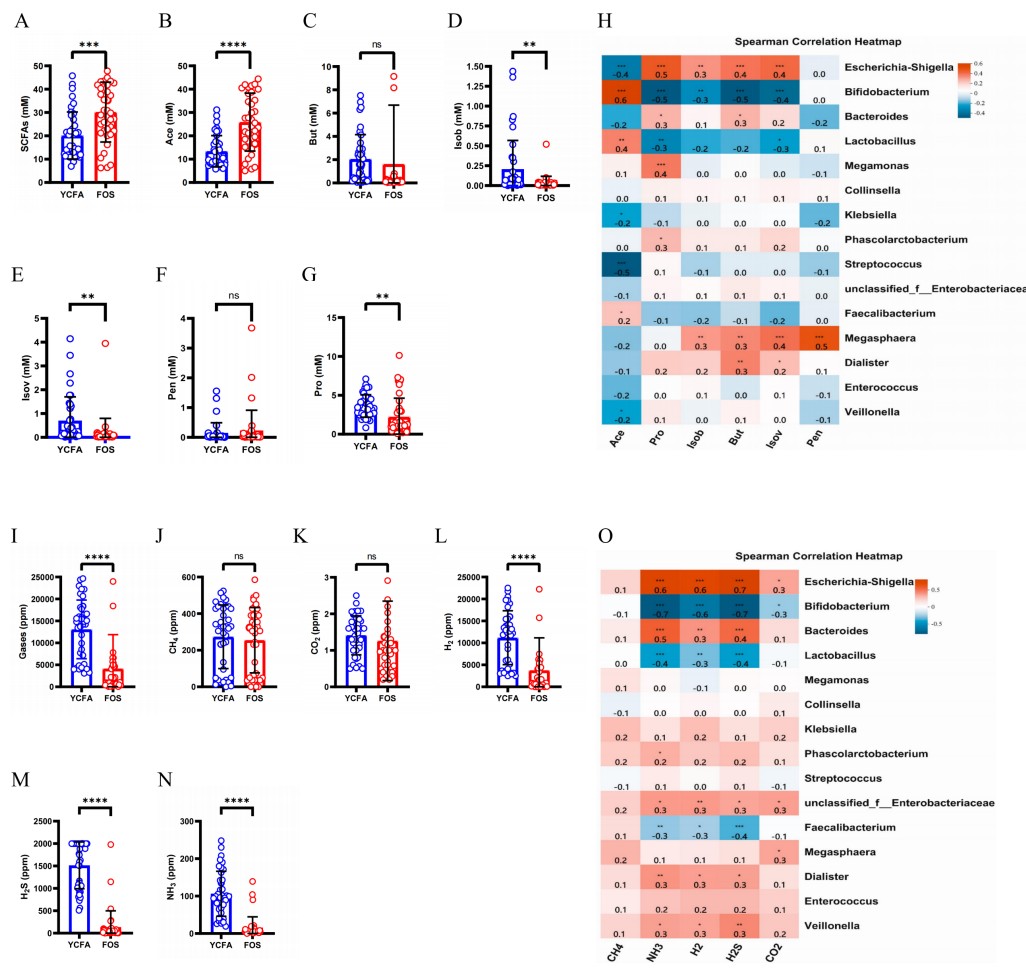

**Figure 3 SCFAs and gas levels in the control group (YCFA group) and the treatment group (FOS group) during *in vitro* fermentation.** (A) Total short-chain fatty acids (SCFAs) levels; (B–G) individual SCFAs levels between the YCFA group and FOS group; (H) correlation heatmap illustrating the relationship between gut microbiota and SCFAs; (I) total gas levels; (J–N) individual gas levels between FOS and YCFA groups; Statistical significance thresholds: (O) correlation heatmap displaying the connection between gut microbiota and gases; statistical significance thresholds: $*p < 0.05$; $**p < 0.01$; $***p < 0.001$; $****p < 0.0001$.

AUC of 0.83 on receiver operating characteristic (ROC) curves (95% CI [0.74–0.92]) (Fig. 4D), confirming their efficacy in distinguishing between the FOS and YCFA groups. The random forest analysis and AUC validation revealed that utilizing both gut microbiota and its metabolites as biomarkers demonstrated higher specificity compared to using gut microbiota alone. Key biomarkers, including $H_2S$, *Bifidobacterium*, $NH_3$, *Parabacteroides*, *Eggerthella*, $H_2$, *Escherichia-Shigella*, Ace, and others, played a crucial role in distinguishing between the FOS and YCFA groups.

## Functional predictive analysis of gut microbiota and metabolites

To investigate the metabolic functions of the gut microbiota, MetOrigin was applied using 16S rRNA sequencing and metabolite data. Metabolites with notable variations between

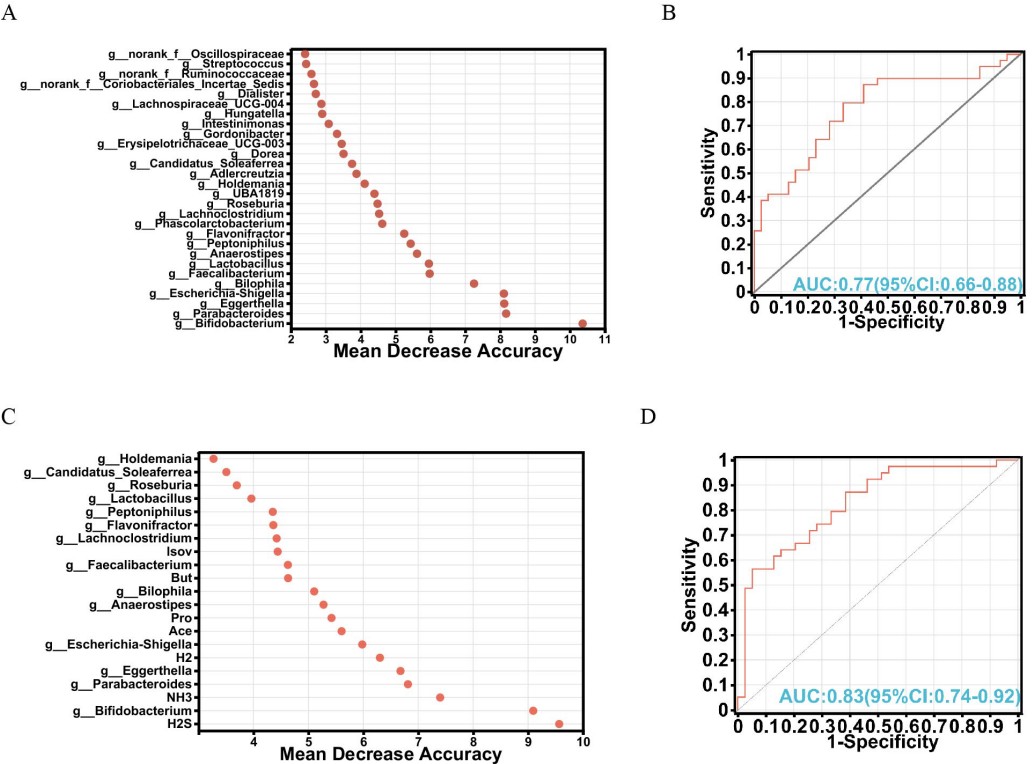

**Figure 4** **Analysis of biomarkers using random forest analysis.** (A) Bubble chart illustrating the variable importance of gut microbiota at the genus level determined through random forest model analysis; (B) evaluation of model candidates through receiver operating characteristic (ROC) analysis of gut microbiota at the genus level; (C) bubble chart depicting the variable importance of short-chain fatty acids (SCFAs), gases and gut microbiota at the genus level using random forest analysis; (D) assessment of model candidates through ROC analysis of gut microbiota at the genus level and metabolites. Statistical significance thresholds: area under the curve (AUC) ≤ 0.5, indicating no diagnostic value; 0.5 < AUC ≤ 0.7, representing low accuracy; 0.7 < AUC ≤ 0.9, indicating a certain degree of accuracy; AUC > 0.9, signifying high accuracy.

the two groups were classified into five categories: foue host-specific metabolites, seven bacterial metabolites, seven drug metabolites, seven food metabolites, and two environment metabolites (Fig. 5A). Through enrichment analysis of metabolic pathways, significant differences in twenty metabolic pathways were identified (Fig. 5B). The metabolic network highlighted noticeable connections ($p < 0.01$) among four metabolites, eight distinct bacteria, and 14 metabolic pathways (Fig. 5C).

## DISCUSSION

In this study, we employed an *in vitro* fermentation to investigate the impact of FOS on gut microbiota and metabolites. This model has gained popularity for its ability to replicate human intestinal conditions while avoiding the ethical and logistical limitations inherent in *in vivo* experiments (*Feng et al., 2023*; *Pi et al., 2024b*; *Wu et al., 2023*). Although *in vitro* models do not account for factors such as intestinal absorption and gastrointestinal

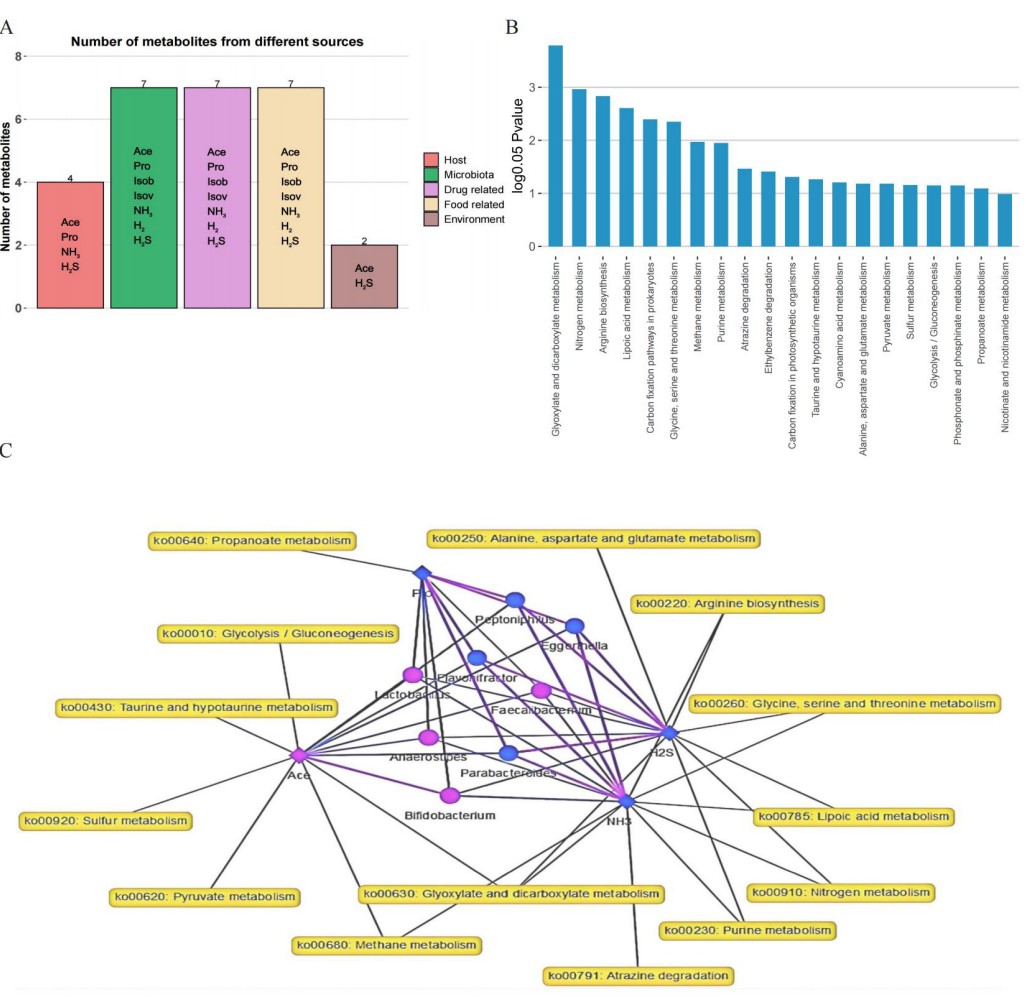

**Figure 5  Analysis of metabolites, microbiota, and metabolic pathways using MetOrigin.** (A) Bar plot illustrating the number of metabolites in different categories; (B) bar plot displaying the enrichment analysis of metabolic pathways; (C) in the network summary of the dietary study on metabolites produced by microbiota, diamond, dot, and rectangle shapes signify correlated metabolites, microbiota, and metabolic pathways, respectively. The purple and blue colors of the nodes denote upregulation and downregulation, respectively. The purple and blue lines indicate positive and negative correlations between microbiota and metabolites, respectively.

secretions, they provide a controlled environment that enables detailed observation of microbial growth and metabolic processes (*Gibson & Fuller, 2000*). By using this approach, we have deepened our understanding of the role FOS plays in modulating gut bacterial communities, which has implications for the management of metabolic disorders and the development of targeted dietary interventions.

The contemporary dietary habits among children have raised considerable concern due to their link with various health problems, notably the escalating prevalence of childhood obesity (*Saavedra, 2022*). Our study on stachyose focused on a relatively lesser-known oligosaccharide highlighting ability to prebiotic effects for childhood obesity. FOS is a

well-known prebiotic that has been extensively researched for its ability to regulate gut microbiota and metabolism (*Mahalak et al., 2022*). However, there is still scope for further investigation into the use of FOS as a dietary fiber supplement to ameliorate childhood obesity in contemporary dietary patterns (*Tandon et al., 2019*). The use of multiple dietary recalls to assess variations in food intake across 24 h and throughout the week adds a new layer of precision to our understanding of dietary patterns. By exploring the effects of FOS as a dietary fiber using an *in vitro* simulated fermentation model, the study seeks to provide valuable insights into the intricate relationship among diet, microbiota, and metabolic outcomes.

The existing literature emphasizes the detrimental effects of contemporary diets on children's health, associating highly processed and sugary foods with the surge in childhood obesity and related metabolic disorders (*Flynn et al., 2022*; *Liberali, Kupek & Assis, 2020*; *Pereira & Oliveira, 2021*). These diets also disrupt the delicate balance of the gut microbiota, leading to a reduction in beneficial bacteria and an increase in potentially harmful ones (*Zhao & Shen, 2023*). The results of our recent study indicate that supplementation with stachyose results in significant alterations in microbioal composition, metabolites, and metabolic pathways in obese children (*Pi et al., 2024a*). These findings suggest that targeting the gut microbiota may represent an efficacious strategy for the management of childhood obesity. In alignment with these findings, this study highlights the influence of dietary components, particularly FOS, on gut microbiota composition and metabolic outcomes in children. Our results revealed no significant difference in microbial $\alpha$-diversity between the FOS and YCFA groups, consistent with existing literature emphasizing the complexity of dietary fiber's impact on overall microbial diversity. While some studies suggest an increase in $\alpha$-diversity with fiber supplementation, our recent study have reported a reduction in $\alpha$-diversity with stachyose supplementation, with variations dependent on the type of fiber and individualized factors (*Holscher, 2017*; *Pi et al., 2024a*; *Tran et al., 2019*; *Zhang et al., 2018*). The observed consistency in microbial diversity challenges the conventional assumption that dietary fiber invariably leads to a substantial change, emphasizing the need for a nuanced understanding of the interactions within the gut ecosystem.

The observed fluctuations in relative abundance illuminate specific genera responding differentially to dietary interventions. Within the FOS group, *Bifidobacterium* stands out as the predominant genus, comprising an impressive 45.12% of the overall relative abundance. This finding aligns with existing literature and our previous research on stachyose, emphasizing the beneficial impact of fiber on the growth of beneficial bacteria, particularly *Bifidobacterium* (*Liu et al., 2017*; *Pi et al., 2024a*; *Zhang et al., 2022b*). The increase in *Bifidobacterium* following FOS consumption holds significant implications, given the well-documented health benefits associated with this genus, including enhanced intestinal health and potential protective effects against pathogenic microbes (*He et al., 2023*; *Vera-Santander et al., 2023*).

LEfSe analysis reveals specific bacterial taxa significantly contributing to differences between FOS and YCFA groups. In the FOS group, notable enrichments include *Bifidobacterium*, *Lactobacillus*, *unclassified Bacilli*, and *Faecalibacterium*. These findings affirm the positive impact of FOS on beneficial bacteria, highlighting the potential of FOS

to create a favorable environment for the proliferation of health-promoting microbial communities (*Cao et al., 2018*; *Lutsiv et al., 2022*; *Van Hul et al., 2020*).

The exploration of SCFAs and gas production offers insights into the metabolic ramifications of dietary interventions, particularly assessing the influence of FOS in contrast to the control group. The cumulative levels of SCFAs demonstrated a noteworthy elevation in the FOS group in comparison to the YCFA group, highlighting the ability of FOS to influence the microbial fermentation process. To analyze the correlation between microbiota and its metabolites, a correlation heatmap was selected, which provides an overall picture of the correlation between microbes and metabolites. The marked rise in cumulative SCFA levels within the FOS group, as opposed to the YCFA group, corresponds with prior literature on stachyose that underscores the role of FOS as fermentable substrates for beneficial bacteria (*Caetano et al., 2016*; *Pi et al., 2024a*). Ace emerged as the predominant contributor to this increase. This discovery aligns with prior research accentuating the role of FOS in fostering the growth of beneficial bacteria, notably *Bifidobacterium* (*Liu et al., 2017*). The notable positive correlation between Ace and *Bifidobacterium* in the correlation heatmap substantiates this relationship, emphasizing the potential role of *Bifidobacterium* in the heightened production of Ace in response to FOS. The correlation heatmap additionally unveils robust negative correlations between Ace and *Streptococcus*, as well as between Pro and *Bifidobacterium*, indicating complex associations between specific SCFAs and microbial taxa.

The analysis of gas composition, encompassing $H_2$, $H_2S$, $CH_4$, and $NH_3$, yields supplementary insights into microbial activity and the metabolic processes triggered by dietary interventions (*Ma et al., 2017*; *Rowland et al., 2018*). The reduced levels of $NH_3$, $H_2$, and $H_2S$ in the FOS group align with the literature suggesting that FOS fermentation reduces gas production. Gut microbiota produces these gases during the fermentation of highly processed and sugary foods, with production influenced by the composition, diversity, and metabolism of the gut microbiota. Certain gases, including $NH_3$ and $H_2S$, are toxic and may elevate the risk of intestinal diseases, type 2 diabetes, obesity, central nervous system diseases, and cardiovascular diseases (*Cai et al., 2022*; *Ma et al., 2017*). The robust positive correlation between $NH_3$ content and *Bacteroides*, along with similar correlations between *Escherichia-Shigella* and the gases $NH_3$, $H_2$, and $H_2S$, underscores the potential involvement of these microbial taxa in gas production. *Escherichia-Shigella*, renowned for its versatile metabolic capabilities, might play a crucial role in gas production through distinct biochemical pathways (*Du et al., 2023*). Conversely, the pronounced negative correlations of $NH_3$, $H_2$, and $H_2S$ with *Bifidobacterium* suggest potential microbial competition and divergence in metabolic pathways among microbial taxa (*Du et al., 2023*; *Gall, 1968*). The noted variations in gas content prompt additional exploration into the microbial ecology influencing gas production. Examining the functional roles of specific microbial taxa, especially *Escherichia-Shigella* and *Bifidobacterium*, in shaping gas profiles could reveal novel insights into microbial community dynamics within the gut.

In our analysis, 21 crucial variables, including diverse gut microbiota and metabolites, were identified as potential biomarkers, achieving an AUC value of 0.83. Key biomarkers, such as $H_2S$, *Bifidobacterium*, $NH_3$, *Parabacteroides*, *Eggerthella*, $H_2$, *Escherichia-Shigella*,

and Ace, played a crucial role in differentiating between the FOS and YCFA groups. Unlike our previous study that used PICRUSt 2 analysis to predict the functional impact of stachyose, the current study employed MetOrigin to explore the metabolic functions of gut microbiota (*Pi et al., 2024a*). Through this approach, metabolites were classified into distinct categories: host-specific, bacterial, drug-related, food-derived, and environmental (*Yu et al., 2022*). Enrichment metabolic pathway analysis revealed significant distinctions in twenty metabolic pathways, offering insights into the broader metabolic alterations triggered by FOS and YCFA consumption. The integrated metabolic network illuminated connections among metabolites, bacteria, and metabolic pathways, providing a visual representation of the intricate relationships within the gut microbiota.

This study has several limitations. *In vitro* simulated fermentation provides a controlled and reproducible setting for examining microbial responses to dietary interventions, but it lacks the complexity and variability of the human gut ecosystem, such as host-microbiota interactions and immune system modulation. Therefore, caution must be exercised in extrapolating our findings to real-life scenarios. However, the controlled nature of the *in vitro* environment allows us to isolate and elucidate specific microbial responses to dietary components, which may be obscured by the complexity of *in vivo* studies. This controlled setting enables us to delve into mechanistic insights, elucidating direct relationships between dietary interventions and microbial outcomes.

Furthermore, this study focuses on the short-term effects, limiting the long-lasting effects of FOS on the composition of gut microbiota and subsequent metabolic outcomes. Understanding the durability of the observed changes in gut microbiota composition and metabolic functions following FOS consumption is crucial for comprehensively evaluating the potential health benefits or risks associated with FOS supplementation. Future research endeavors could address this limitation by incorporating longitudinal study designs, with follow-up assessments conducted over an extended period.

The potential limitation of this study is the absence of a control group of healthy volunteers. The rationale for the decision not to include a healthy control group is as follows: (1) Given the already well-established differences in gut microbiota between healthy and obese populations, our decision to focus on the obese cohort allowed us to concentrate on the unique microbiota shifts within this group without the need for direct comparisons to a healthy control group. (2) To investigate the impact of FOS on the gut microbiota of obese children was the primary purpose of our study. Given this specific focus, we sought to maintain homogeneity within the cohort by limiting the study to children with obesity. The inclusion of a healthy control group, while valuable in some contexts, could introduce additional variability related to baseline differences in gut microbiota composition, metabolic profiles, and dietary habits between obese and healthy children. Our goal was to isolate the impact of FOS within the obese population, as this group is of particular interest in the context of microbiota-targeted interventions.

The relatively small sample size likely constrained the statistical power to discern significant gender-related differences. The inclusion of larger, more diverse sample sizes in future research would facilitate the elucidation of whether gender exerts an influence on responses to FOS treatment. Additionally, the study's emphasis on specific

dietary components, like FOS, may neglect the broader dietary context, constraining the comprehensive understanding of the intricate interactions within the gut. Including an additional experimental group supplemented with a different type of dietary fiber, such as inulin or resistant starch, would provide valuable context for interpreting the effects of FOS and elucidating its unique contributions to gut health.

Moreover, the absence of consideration for individual differences in gut microbiota composition represents a notable limitation. Gut microbiota are highly personalized and can vary significantly among individuals due to factors such as metabolic disease, genetics, lifestyle, and dietary habits. In this study, obese children were specifically selected; however, participants with a clinical diagnosis of diabetes, metabolic syndrome, or other chronic conditions were not excluded. Proper consideration of these individual differences is essential for elucidating the nuanced effects of FOS supplementation and for tailoring dietary interventions to individual needs. Future studies should incorporate comprehensive profiling of participants' gut microbiota at baseline to account for individual variation and explore potential interactions between host factors, dietary habits, and FOS supplementation.

## CONCLUSIONS

This study explored the impact of FOS on gut microbiota and metabolites within the context of current dietary patterns and obesity in children, using an *in vitro* simulated fermentation process. The results revealed significant alterations in both the diversity and composition of the gut microbiota post-FOS treatment. Specifically, there was a noteworthy increase in beneficial bacteria such as *Bifidobacterium* and *Lactobacillus*, accompanied by a reduction in harmful bacteria like *Escherichia-Shigella* and *Bacteroides*. Moreover, a substantial increase in SCFAs, especially Ace, was observed, coupled with a significant decline in gas production, encompassing $NH_3$, $H_2$, and $H_2S$. Key biomarkers indicative of the FOS intervention's efficacy were identified, including $H_2S$, *Bifidobacterium*, $NH_3$, *Parabacteroides*, *Eggerthella*, $H_2$, *Escherichia-Shigella*, and Ace. These findings, acquired through *in vitro* simulated fermentation, provide a valuable understanding of the complex interplay among contemporary dietary patterns, gut microbiota, and microbial metabolites in children. Nevertheless, it's crucial to acknowledge that *in vitro* conditions may not fully replicate the complexities of the real gut environment, further clinical studies are essential to explore long-term effects and consider additional dietary factors for a more comprehensive understanding.

### Funding

This research was supported by the Zhejiang Provincial Natural Science Foundation of China under Grant No. YY22H280002. The funders had no role in study design, data collection and analysis, decision to publish, or preparation of the manuscript.

## Grant Disclosures

The following grant information was disclosed by the authors:
Zhejiang Provincial Natural Science Foundation of China: YY22H280002.

## Competing Interests

The authors declare there are no competing interests.

## Author Contributions

- CanFeng Mo conceived and designed the experiments, analyzed the data, prepared figures and/or tables, and approved the final draft.
- Shan Zhou performed the experiments, authored or reviewed drafts of the article, and approved the final draft.
- Zhi Du analyzed the data, prepared figures and/or tables, and approved the final draft.
- Xiangxin Huang analyzed the data, authored or reviewed drafts of the article, and approved the final draft.

## Human Ethics

The following information was supplied relating to ethical approvals (i.e., approving body and any reference numbers):

Ethical approval for the study protocol was obtained from the Ethical Committee of Hangzhou Centers for Disease Control and Prevention (No. 202047).

## Data Availability

Sequencing data is available at the Sequence Read Archive (SRA): PRJNA1201739.

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
