# Peer review of "Impact of fructooligosaccharides on gut microbiota composition and metabolite production: implications for childhood obesity"

_PeerJ, doi:10.7717/peerj.19894_

## Round 0.1 · original submission · Major Revisions

·

Basic reporting

There were no figure legends described anywhere in the manuscript or figure.

Some numbers (correlation coefficients) in the heatmap in Figure 3 H and O are difficult to see as they have been masked by the dark blue color of the correlation.

In Figure 5C, the relationship between each nodes was depicted using dark green and dark red, which is hard to differentiate and not color-blind friendly.

In Figure 5A, it would be better to show what metabolites are in each category (host, microbiota-derived etc.) by the method.

Experimental design

Methods for the analysis was not described with sufficient detail:
- How were the reads treated prior to alpha diversity analysis? Was there any rarefaction? If so, at what sampling depth?
- What are the statistical tests used to assess the significance of groups for alpha and beta diversity?
- Was there any relative abundance threshold cut-off for beta diversity and differential abundance analysis? Were the counts rarefied, transformed into relative abundance or used as is for those analysis?
- Again, how was the data treated prior to random forest analysis?
- It was unclear whether isolation of gut microbiota was done under anaerobic conditions or not (line 120-124). If not, that is a serious flaw of experimentation as microbiome composition would have been skewed even before experimentation.
- It was not mentioned in the methods section that the authors have conducted metabolic pathway enrichment analysis, but the figures were presented in the results section.

Validity of the findings

no comment

Additional comments

There were many basic information missing from the methods section in the abstract section, including:
- how long was the in-vitro fermentation
- was it a single or multiple timepoint study?
- sequencing method to investigate microbiome composition: 16S amplicon sequencing or shotgun metagenomics?

Were there other comorbidities in these obese children? Diabetes or other metabolic syndromes are typically found in obese patients, and perhaps medication was also used to treat those syndromes. Any of those would have a significant effect on the gut microbiome. Thus, it was difficult to assess potential underlying differences even prior to experimentation. The authors also did not show whether there were any microbiome features that are associated with clinical factors such as: gender, age, BMI, medication, even after fermentation. Even assessing the potential role of these factors in FOS-fermentation would be interesting.

Reviewer 2 ·

Basic reporting

This study investigated the effect of FOS on gut microbiota and metabolites in context with current dietary patterns and obesity in children. The investigators used an in vitro simulated fermentation process to address their study objective.
It is an interesting study on an important issue-how prebiotics can be used to modify gut microbiome and resultant metabolom.
This is a well designed study and generated important relevant information which will be useful as reference in future clinical studies.
The paper needs minor editing / corrections prior to its acceptance for publication.

Experimental design

No comment

Validity of the findings

Findings are valid and vastly improves our current understanding.
Among the SCFA produced, the status of butyrate is not discussed, although isobutyrate has been discussed.

Additional comments

Review
Impact of fructooligosaccharfides on gut microbiota composition and metabolite production: Implications for childhood obesity (#112515)

This study investigated the effect of FOS on gut microbiota and metabolites in context with current dietary patterns and obesity in children. The investigators used an in vitro simulated fermentation process to address their study objective.

It is an interesting study on an important issue-how prebiotics can be used to modify gut microbiome and resultant metabolom.

This is a well designed study and generated important relevant information which will be useful as reference in future clinical studies.

The paper needs minor editing / corrections prior to its acceptance for publication.

COMMENTS:
Structure and Criteria:
Please read the 'Structure and Criteria' page for guidance. Done

Custom checks:
Make sure you include the custom checks shown below, in your review.
Raw data check Review the raw data. Checked

Image check If this article is published your review will be made public. Checked

You can choose whether to sign your review. If uploading a PDF please remove any identifiable information (if you want to remain anonymous). Check that figures and images have not been inappropriately manipulated. Checked

English correction needed. Meaning of a few sentences are not clear
For example
Lines 275-276
Our study on Stachyose focused on a relatively lesser-known oligosaccharide with, highlighting its
Comment: Corrected version can be:
Our study on Stachyose focused on a relatively lesser-known oligosaccharide highlighting
ability to prebiotic effects for childhood obesity.

Line 280 (typographic effort).
(Tandon et al. 2019)c.
Comment: c is not needed here.

Result:
SCFAs and Gas Production During Fermentation.
Comment: The status of butyrate, an important SCFA is not mentioned

Discussion:
Line 338-339: The reduced levels of NH3, H2, and H2S in 339 the FOS group align with the literature suggesting that FOS fermentation inhibits gas production.
Comment: Strong statement. It can be changed to “The reduced levels of NH3, H2, and H2S in 339 the FOS group align with the literature suggesting that FOS fermentation reduces gas production.”

References
References are not in a single format. Reference # 2 and # 3 are not in the same format.

In # 2, the full name of the journal is written, whereas in the # 3, the name of the journal is written in abbreviated form. There are few other such examples

#2 Benito I, Encio IJ, Milagro FI, Alfaro M, Martinez-Penuela A, Barajas M, and Marzo F. 2021. 446 Microencapsulated Bifidobacterium bifidum and Lactobacillus gasseri in Combination 447 448 with Quercetin Inhibit Colorectal Cancer Development in Apc(Min/+) Mice. International Journal of Molecular Sciences 22. 10.3390/ijms22094906

# 3 Bi Z, Cui E, Yao Y, Chang X, Wang X, Zhang Y, Xu GX, Zhuang H, and Hua ZC. 2022. 450 Recombinant Bifidobacterium longum Carrying Endostatin Protein Alleviates Dextran 451 452 Sodium Sulfate-Induced Colitis and Colon Cancer in Rats. Front Microbiol 13:927277. 10.3389/fmicb.2022.927277


Ashfaque Hossain
Consultant Microbiologist
Atlanta, GA 30033, USA
[email protected]
001-402-830-1150

---

## Round 0.2 · Minor Revisions

The reviewer was happy that the majority of their previous comments had been addressed. They have highlighted one minor issue (see below), which once fixed will allow the manuscript to proceed to acceptance.

·

Basic reporting

-

Experimental design

The authors have very much improved the overall manuscript, with more details on specific aspects of the experimental method. There is just one small piece of conflicting information in the manuscript:
Line 163-165: The authors stated that the reads were rarefied to 54,000 prior to alpha and beta diversity analysis.

However,
On Lines 203-205: The authors stated that the count data was converted to relative abundance prior to beta diversity testing.

Does it mean that the reads were rarefied and then scaled to relative abundance for beta diversity analysis? This could have implications for the interpretation of results.

FYI: The detail on the PCR instrument and program is not needed in the manuscript and can be deleted.

Validity of the findings

The conclusions of the study are aligned with the results.

---

## Round 0.3 · accepted · Accept

Thank you for clarifying that last remaining point. The manuscript is now ready for publication.